# Real-World Outcomes of Immunotherapy in Second- or Later-Line Non-Small Cell Lung Cancer with Actionable Genetic Alterations

**DOI:** 10.3390/cancers15225450

**Published:** 2023-11-16

**Authors:** Soojin Jun, Sehhoon Park, Jong-Mu Sun, Se-Hoon Lee, Jin Seok Ahn, Myung-Ju Ahn, Juhee Cho, Hyun Ae Jung

**Affiliations:** 1Department of Clinical Research Design and Evaluation, Samsung Advanced Institute for Health Sciences & Technology (SAIHST), Sungkyunkwan University, Seoul 06355, Republic of Korea; ddori2@g.skku.edu (S.J.); jcho@skku.edu (J.C.); 2Division of Hematology-Oncology, Department of Medicine, Samsung Medical Center, Sungkyunkwan University School of Medicine, Seoul 06351, Republic of Korea; sehhon.park@samsung.com (S.P.); jongmu.sun@skku.edu (J.-M.S.); sehoon.lee@samsung.com (S.-H.L.); ajis@skku.edu (J.S.A.); silkahn@skku.edu (M.-J.A.)

**Keywords:** non-small cell lung cancer, immune checkpoint inhibitors, actionable genetic alterations, predictive biomarkers

## Abstract

**Simple Summary:**

This study focused on actionable genetic alterations (AGAs) subtypes and pathologic or genetic biomarkers influencing the efficacy of immune checkpoint inhibitor (ICI) therapy in a real-world setting. In *advanced* non-small cell lung cancer (NSCLC) patients, the response to ICI monotherapy varies among AGAs. In previous studies, while patients with *KRAS*, *BRAF*, and *MET* exhibited favorable efficacy, it did not appear in patients with *EGFR*, *ALK*, *ROS1*, or *RET*. In this study, ICI monotherapy benefits differed across AGA subtypes but reaffirmed that *KRAS*, *MET*, and *BRAF* patients experienced longer benefits in the second- or later-line therapy. PD-L1 was a positive predictive biomarker, but not TMB. Co-existing *STK11* with *KRAS* and *TP53* with *MET* mutation were negatively correlated with ICI responses. Despite the limitation of a small sample size for some rare mutations, this study can still provide valuable insights that may guide clinical decision making and further research to validate the findings.

**Abstract:**

Introduction: While the efficacy of immune checkpoint inhibitors (ICIs) in treating non-small cell lung cancer (NSCLC) patients with actionable genetic alterations (AGAs) is modest, certain patients demonstrate improved survival. Thus, this study aimed to evaluate the benefits of ICIs in NSCLC patients with diverse AGAs and verify the predictive biomarkers of ICI efficacy. Methods: From January 2018 to July 2022, this study compared the progression-free survival (PFS) of NSCLC patients with different AGAs treated with ICI monotherapy as second- or later-line therapy at Samsung Medical Center. To ascertain the predictors of ICIs efficacy, we adjusted ICIs’ effects on PFS in terms of clinical and molecular biomarkers. Results: *EGFR* (46.0%) was the most prevalent mutation in 324 patients. In multivariate analysis, PD-L1 positivity (tumor proportion score (TPS) ≥ 1%) (HR = 0.41) and the use of steroids for immune-related adverse events (HR = 0.46) were positive factors for ICI therapy in the AGAs group. Co-existing mutation of *STK11* with *KRAS* mutation (HR = 4.53) and *TP53* with *MET* mutation (HR = 9.78) was negatively associated with survival. Conclusions: The efficacy of ICI treatment varied across AGA subtypes, but patients with *KRAS*, *MET*, and *BRAF* mutations demonstrated relatively long-duration benefits of ICI therapy. PD-L1 was a significant positive predictive biomarker in all AGA groups.

## 1. Introduction

Immune checkpoint inhibitors (ICIs) block the pathways through which cancer cells evade the immune system [1]. While the utilization of ICIs has significantly improved survival in patients with advanced NSCLC, room remains for improvement. For instance, the overall response rate (ORR) is approximately 20% for monotherapy; not all patients benefit from ICIs [2]. Thus, many studies have been undertaken to identify biomarkers to optimize predictive biomarkers for the efficacy of ICI. The positive predictive nature of programmed death-ligand 1 (PD-L1) and tumor mutational burden (TMB) for ICI therapy is well established [3,4,5,6]. Additional molecular or clinical biomarkers, such as tumor-infiltrating lymphocytes (TILs), gene expression profiling (GEP), mismatch repair and microsatellite instability, neutrophil-to-leukocyte ratio, somatic mutations including actionable genetic alterations (AGA) or co-existing mutations of *STK11*, *KEAP1*, and *TP53* with other driver mutations [7,8], smoking history, antibiotics, microbiome, and the occurrence of immune-related adverse event (irAE) further inform clinical decision-making [2,9].

While ICIs have demonstrated benefits in subsets of AGAs, their efficacy can be influenced by characteristics of the tumor microenvironment (TME), which consists of various components, including PD-L1 and neoantigens, and varies with different types of AGAs [9]. Because ICIs show modest benefits in patients with AGAs, the NCCN guideline [10] recommends treating ICIs after tyrosine kinase inhibitor therapy failure, as a second- or later-line treatment [11]. Furthermore, some prospective observational studies report a lower response rate to ICI treatment in patients with co-existing mutations of *KRAS*, *STK11*, and *TP53* than patients with wild-type genes [7,8]. However, previous observations of efficacy and biomarkers for various AGA variances in non-small cell lung cancer (NSCLC) were insufficient because patients with AGAs have been ineligible for clinical trials and only specific factors, such as the PD-L1 and TMB, were available rather than comprehensive molecular data [7,12].

Meanwhile, cancer incidence and outcomes exhibit substantial disparities among racial and ethnic groups, with differing levels of exposure to risk factors and impeding access to high-quality cancer prevention, early detection, and treatment [13,14]. Since 2017, a genetic panel test utilizing next-generation sequencing (NGS) technology has been officially designated as part of the national health insurance coverage for lung cancer patients in South Korea, and this test expedites the identification of genetic mutations. Consequently, South Korea is a conducive environment for implementing precision medicine based on precise and abundant NGS data for patients in advanced NSCLC [15]. As a result, we aimed to evaluate outcomes of immunotherapy in second- or later-line non-small cell lung cancer, segmented by AGA, using a large hospital registry data.

## 2. Methods

### 2.1. Study Design and Study Population

This retrospective cohort study was conducted using a oncology data registry at Samsung Medical Center (SMC), referred to as Real-time autOmatically updated data warehOuse in healThcare (ROOT) [16]. We obtained the data from January 2018 to July 2022 for this study. Inclusion criteria were advanced or metastatic NSCLC patients who received ICI therapy and had tested for AGAs. Patients who were treated with ICI as first-line treatment or combination therapy, and those who had a follow-up duration of less than 1 month for PFS, were excluded from the study.

The study was conducted pursuant to the Declaration of Helsinki. The Institutional Review Board of SMC granted approval for this study (IRB no. 2022-08-013). The IRB waived the requirement for informed consent due to the retrospective nature of this study.

### 2.2. Outcomes

The primary endpoint was progress free survival (PFS). Overall survival (OS), ORR, and the 12-month PFS rate were the secondary endpoints. PFS was defined as the time from the start of ICI treatment to the documentation of disease progression or death from any cause. OS was calculated as the time from ICI treatment initiation to death from any cause. ORR was a measure of how many patients achieved either a complete response (CR) or a partial response (PR) to ICI therapy. The response was evaluated according to the Response Evaluation Criteria in Solid Tumors (RECIST) version 1.1.

### 2.3. Identification of Actionable Genetic Alteration, PD-L1, and TMB

AGAs were identified using companion diagnostics for each genetic alteration, such as NGS, polymerase chain reaction, immunohistochemical stain, and fluorescence in situ hybridization. The number and rate of each driver mutation tested using NGS are shown in Appendix A.

The AGAs included in this study were as follows: *BRAF* V600E mutations; *EGFR* mutation (exon 19 deletion, L858R mutation in exon 21, exon 20 insertion); *HER2* (ERBB2) mutation; *KRAS*, mutations including G12A, G12C, G12D, G12V, G12R, G12F, G13D, and others; *MET* exon 14 skipping mutations and *MET* amplification; *ALK* rearrangement; *ROS1* rearrangement; *RET* mutation or rearrangement; and NTRK rearrangement. A comprehensive genomic test was performed in an accredited reference laboratory with validated methods using the CancerSCAN panel (GENINUS, https://www.kr-geninus.com (accessed on 12 November 2023), Seoul, Republic of Korea) and TruSight Oncology 500 assay (Illumina, https://www.illumina.com (accessed on 12 November 2023), San Diego, CA, USA).

PD-L1 expression was examined by using anti-PD-L1 antibodies (22C3, SP263, and SP142) and PD-L1 high was defined as ≥1%. TMB was defined as the inclusion of all non-synonymous and synonymous mutations, excluding germline variants and driver mutations. TMB-high was determined as ≥10 mutations/mega base (mut/Mb). Additionally, co-existing mutations of *TP53*, *STK11*, and *KEAP1* were examined.

In addition, clinical data (sex; age at diagnosis; smoking history; Eastern Cooperative Oncology Group (ECOG) performance status; histologic finding; underlying co-morbidity, such as hypertension (HTN) and diabetes mellitus (DM); and concomitant medications during ICI treatment, such as radiotherapy, steroids, and antibiotics) were obtained from the ROOT.

### 2.4. Statistical Analyses

For quantitative variables, the summary is provided in terms of medians and 95% confidence intervals (CIs). Categorical variables, on the other hand, are summarized using numbers and percentages. Between-group comparisons for categorical variables were conducted using either the chi-squared test or Fisher’s exact test to analyze the differences. All *p* values were two-sided and CIs were 95%, with statistical significance set at *p* < 0.05. The Kaplan–Meier method was applied to estimate median PFS (mPFS) and median OS (mOS), and the log-rank test was used to compare differences in event-time distributions among the oncogenic driver subgroups. Patients who did not report any event were censored at the start date of a new therapy or at the last follow-up date for PFS and at the last follow-up date for OS. Cox proportional hazards regression models were used to calculate hazard ratios (HRs). The multivariate analysis considered several important clinical and molecular factors as significant variables. The Cox proportional hazards model assumption was inspected using Schoenfeld residuals against the transformed time. R version 4.2.2 was utilized for conducting all statistical analyses.

## 3. Results

### 3.1. Clinical Characteristics

There were 324 and 602 patients with and without AGAs who received ICI monotherapy as second- or later-line therapy at SMC during the study period (Appendix A). Among AGA subtypes, *EGFR* mutation was the most common (46%, n = 149), followed by *KRAS* (22.2%, n = 72), *HER2* (10.5%, n = 34), *MET* (9.9%, n = 32), *ALK* (3.7%, n = 12), *BRAF* (2.8%, n = 9), *ROS1* (2.8%, n = 9), and *RET* (2.2%, n = 7) (Table 1). No NTRK mutation was detected. Regarding the clinical characteristics, there were no significant differences except in sex (*p* = 0.01) smoking status (*p* = 0.002), and line of therapy (*p* < 0.001) across the AGA subtypes. In total, 46.9% (n = 152) of the patients were female and 46.3% (n = 150) were ex- or current smokers, and 47.2% (n = 153) were identified as patients who received ICI in the 2nd line therapy in the AGA group (Appendix A). The group with *MET* alterations had the highest proportion of smokers (65.6%, n = 21), whereas the group with *RET* alterations had no smokers. Steroid use owing to immune-related adverse events (irAE) and antibiotics use were found in 23.5% (n = 76) and 15.7% (n = 51) of the AGA group, respectively (Appendix A).

### 3.2. Molecular Biomarkers (PD-L1, TMB, and Co-Existing Mutations)

Among 324 patients with AGAs, PD-L1 expression and TMB status data were available for 317 (97.8%) and 92 (28.4%) patients, respectively. The median PD-L1 tumor proportion score (TPS) varied with the type of AGA as follows: 50% in the *BRAF* (n = 9) and *ROS1* (n = 9); 25% in *ALK* (n = 11); 10% in the *EGFR* (n = 145), *MET* (n = 31), and *RET* (n = 7); 7.5% in the *KRAS* (n = 72); and 0 in *HER2* (n = 34) subgroups (Table 2). In addition, 77.8% of the patients with *BRAF* and *ROS1* mutations were PD-L1 TPS-positive (PD-L1 ≥ 1%) (Table 2 and Figure 1A).

In general, the median TMB value was lower in AGAs compared to the wild type (7.0 muts/Mb vs. 10.5 muts/Mb) (Table 2). The distribution of TMB according to each type of AGA is shown in Figure 1B. High TMB (≥10 muts/Mb) was observed in 68.8% of patients with *HER2* mutations, whereas none of the patients with *ALK*, *BRAF*, and *ROS1* mutations had high TMB, which was statistically significant across all subtypes (*p* < 0.01) (Table 2).

In terms of co-existing mutations, *TP53* was the most prevalent (63.1%) mutation and with a frequency of occurrence with *EGFR* (75.5%) and *MET* (73.9%) subtypes. Frequency of occurrence of *STK11* was highest with *KRAS* (16.7%), followed by *BRAF* (14.3%). The occurrence of *KEAP1* mutation was most common with *HER2* (12.1%), while none of the patients in the *ALK*, *BRAF*, *ROS1*, or *RET* subgroups experienced occurrence of co-existing mutations (Table 2 and Figure 1C).

### 3.3. Treatment Outcomes of Immune Checkpoint Inhibitors

The 324 patients with AGAs had significantly shorter mPFS periods than the 602 wild-type patients (2.0 months; 95% CI, 2.0–2.0 vs. 2.1 months; 95% CI, 2.0–3.0, *p* < 0.001). The mPFS for each type of AGA was as follows: 3.1 months (95% CI, 2.0–10.1) for *MET*, 3.0 months (95% CI, 2.0–NR) for *ROS1*, 2.1 months (95% CI, 2.0–3.1) for *KRAS*, 2.0 months (95% CI, 2.0–3.0) for *HER2*, 2.0 months (95% CI, 2.0–NR) for *ALK*, 2.0 months (95% CI, 2.0–NR) for *BRAF*, and 2.0 months (95% CI, 1.0–NR) for *RET*. The shortest mPFS was observed with *EGFR* mutation (2.0 months; 95% CI, 2.0–2.0) (Table 3 and Figure 2A). Notably, patients with *KRAS*, *MET*, and *ROS1* mutations had comparable PFS to those with wild-type genes (Table 3).

Additionally, the ORR (CR, n = 1 [0.3%]; PR, n = 44 [13.6%]) of patients with AGAs was lower than that of wild-type patients (CR, n = 6 [1.0%]; PR, n = 128 [21.3%]), but the difference was not significant (13.9% vs. 22.3%, *p* = 0.82). A subgroup analysis showed that one CR was observed with *ROS1* mutation. Among AGAs, the highest ORR was with *RET* (28.6%), followed by *MET* (25.0%), *KRAS* (22.2%), and *ROS1* (22.2%) (Table 3). Based on the 12-month PFS rates, the *MET* (23.5%), *BRAF* (22.2%), and *KRAS* (17.4%) subgroups had better long-term response rates than the *EGFR* (6.4%) and *ALK* (0%) subgroups (Table 3). Compared to the other subgroups, *EGFR* and *ALK* subgroups benefited less from the ICIs treatment.

Among the AGA subgroups, the analysis of the impact of PD-L1 level on immunotherapy identified a significant difference in mPFS between high (≥1%) and low (<1%) (4.0 months; 95% CI 2.1–6.1 vs. 2.0 month; 95% CI 1.0–3.0, *p* < 0.001) in the *KRAS* subgroup (Appendix A).

Subgroup analysis was performed to explore the heterogeneity of each AGA type. In the *EGFR* subtype, mPFS was lower for patients with T790M (n = 41) than those without T790M (n = 108) (2.0 months; 95% CI 1.0–2.0 vs. 2.0 months; 95% CI 2.0–2.1, *p* = 0.047) (Figure 2B). In *KRAS*-mutant subtypes, G12V (n = 19, 27%) was the most prevalent subtype, followed by G12C (n = 16, 22%) and G12D (n = 16, 22%) (Appendix A), but no difference was observed in mPFS per each subtype (*p* = 0.23) (Appendix A). Moreover, there were no differences in mPFS between G12C (n = 16, 22.2%) and non-G12C (n = 56, 77.8%) in the *KRAS* subtype (3.0 months; 95% CI 2.0–NR vs. 2.0 months; 95% CI 2.0–3.1, *p* = 0.95) (Figure 2C). In the *MET*-mutant subtypes, no difference was found between exon 14 skipping (n = 5, 15.6%) and amplification (n = 27, 84.4%) (mPFS; NR; 95% CI 5.1–NR vs. 3.0 months; 95% CI 2.0–7.1, *p* = 0.10) (Figure 2D).

### 3.4. Clinical and Molecular Predictive Markers for Immune Checkpoint Inhibitor

The potential predictive markers varied between AGA subtypes in the multivariate analysis (Appendix A). PD-L1 positivity (PD-L1 ≥ 1%) (HR = 0.41; 95% CI, 0.25–0.68; *p* < 0.001) and the use of steroids for irAEs (HR = 0.46; 95% CI, 0.23–0.92; *p* = 0.03) were the positive predictive factors associated with longer PFS in the AGA group (Figure 3A).

In patients with *KRAS* mutations, PD-L1 positivity (≥1%) was associated with higher survival with ICI treatment (HR = 0.21; 95% CI, 0.07–0.64; *p* = 0.01). In contrast, co-existing mutation with *STK11* (HR = 4.53; 95% CI, 1.05–19.47; *p* = 0.04) negatively affected survival (Figure 3B). In the *EGFR* mutation group, no predictive biomarkers were found (Figure 3C). In the *MET* alteration group, PD-L1 positivity (≥ 1%) had a positive effect on ICI treatment compared to PD-L1 negativity (HR = 0.17; 95% CI, 0.03–0.89; *p* = 0.04), whereas *TP53* co-existing mutation negatively impacted mPFS with ICI treatment (HR = 9.78; 95% CI, 1.22–78.11; *p* = 0.03) (Figure 3D). In patients with *HER2* mutations, univariate analysis revealed that steroid use to treat irAE was positively associated with ICI treatment (HR = 0.38; 95% CI, 0.15–0.95; *p* = 0.04); however, this significance was lost in the multivariate analysis (HR = 0.46; 95% CI, 0.07–3.0; *p* = 0.42). Moreover, smoking, antibiotic use during ICI treatment, TMB levels, and *KEAP1* mutations did not demonstrate statistical significance as predictive biomarkers of ICI treatment efficacy (Appendix A). In the *BRAF-*, *ALK-*, *ROS1*-, and *RET*-mutant subgroups, univariate analysis was performed although several factors could not be calculable owing to the small sample size, and no predictive biomarker was found (Appendix A). In Figure 4, the graphical algorithm depicted favorable and unfavorable predictive molecular findings, presented based on multivariate analysis, for the AGA group, *KRAS* and *MET* subtypes.

## 4. Discussion

This study demonstrated that outcomes of ICIs treatment varied according to the type of AGA. Similarly to a previous study, our study showed that *KRAS*, *MET*, and *BRAF* subgroups benefited more from ICIs treatment compared to *EGFR* and *ALK* subgroups [17]. In the IMMUNOTARGET study, ICI treatment had favorable effects in patients with *KRAS*, *BRAF*, and *MET* mutations but not in those with *EGFR*, *ALK*, *ROS1*, or *RET* mutations. Specifically, the 12-month PFS indicated that long-term responders were more prevalent in *KRAS* (25.6%), *MET* (23.4%), and *BRAF* (18.0%) subgroups than in *RET* (7.0%), *EGFR* (6.4%), and *ALK* (5.9%) subgroups [17].

The immunogenicity of *BRAF* and *KRAS* mutations has been studied in vivo and in vitro [18,19]. By integrating the examination of immune-related scores (such as GEP scores, T cell markers, IFN-γ signatures, and chemokines) into our findings, we could deduce that *BRAF* mutation might lead to equilibrated immunomodulatory effects [18]. The presence of a *KRAS* mutation exhibited a correlation with an inflammatory TME and tumor immunogenicity. Indeed, *KRAS* mutation is associated with a higher proportion of PD-L1+/TIL + cells, indicating *KRAS*-mutant tumors exhibit an inflammatory phenotype characterized through adaptive immune resistance. Additionally, *KRAS* mutation promotes elevated TMB and enhanced immunogenicity [19].

The existence of different ICIs treatment outcomes for each *KRAS* subtype (G12C vs. non-G12C) is controversial. Our study and Jeanson et al.’s study showed that there was no difference between the group with G12C and the group with non-G12C subtype [20]. On the contrary, in Taiwan Wu et al. reported that the G12C subtype was favorably associated with ICI effectiveness compared to G12V [21].

A meta-analysis reported that patients with *EGFR* mutations did not benefit from ICI monotherapy as a second-line therapy compared to docetaxel [9]. Consistent with prior findings [22,23,24], the *EGFR* subtype in the present study did not benefit from ICI treatment in terms of mPFS across all subgroups, even for the high-TPS group (≥50%).

Consistent with prior findings [17,25], we demonstrated that high PD-L1 expression predicts prolonged mPFS in the *KRAS* and *MET*-mutant subgroups, as well as the whole AGA group treated with ICI. In NSCLC, smoking is associated with higher TMB, which potentiates ICI therapy [11]. In our study, however, both high TMB and smoking status did not influence the survival of any AGA group. Notably, the variability of the median TMB per AGA subgroup in this study was consistent with the systematic review by Sha et al. [26].

The frequency of co-existing mutation with *STK11* also varied across AGAs in our study. Additionally, the prevalence rate of 16.7% for *STK11* and *KRAS* co-existing mutation in our study was less than the 35% reported by a retrospective study (STRIKE registry-CLICaP) among Hispanics (n = 13) [7]. Based on prospective VISION and MAGIC I studies by Pavan et al., *STK11* with *KRAS* mutations was associated with poor ICI outcomes [8,27]. In line with the result, the *STK11*-*KRAS* co-existing mutation could be a potential biomarker associated with severe ICI outcomes. Furthermore, the presence of the *TP53* mutation was found to correlate with declining PFS with ICI treatment in the *MET* subgroup, which is consistent with the trial reported by Pavan et al. [8]. The potential predictive role of *TP53* in NSCLC is still controversial [28] as both *MET* and *TP53* directly contribute to regulating PD-L1 [25]. Nevertheless, further clinical trials are necessary to investigator this factor in the future.

Additional findings of the present study include the revelation that steroid use to treat irAE during ICI therapy is associated with prolonged mPFS in the AGA group, meaning steroid use to treat irAE does not negatively impact PFS. This result is consistent with a meta-analysis [29] that concluded that irAEs are beneficial for survival and response in advanced NSCLC. Moreover, if steroids were introduced during the initial eight weeks of ICI therapy in patients with NSCLC without any indication of cancer, there was no adverse effect on the prognosis [30].

Several limitations were encountered in this study. First, it relied on a retrospective analysis of an observational study. AGAs were detected using diverse testing methods, and TMB data were only available for 28.4% of the total AGA patient. Second, analyses regarding rare alterations such as *BRAF* and *ROS1* were limited. Finally, these data were extracted from a single center, making it difficult to generalize the findings. Despite these limitations, our study reflects real-world clinical practice and provides valuable insights regarding the utilization of ICI monotherapy in patients with advanced or recurrent NSCLC harboring AGAs and suggests the need for personalized treatment strategies based on genomic profiling. Further large-scale prospective trials are required to validate these biomarkers and to explore new predictive biomarkers for effectiveness of ICI treatment.

## 5. Conclusions

This study explored the differences in the efficacy of ICI therapy for NSCLC patients with AGAs who received ICI monotherapy as a second- or later-line treatment and the potential predictive biomarkers that influence these responses. PD-L1 was a significant positive predictive biomarker in the AGA group. The overall response and long-term efficacy of ICI treatment varied across AGA subtypes; however, patients with *KRAS* and *MET* mutations experienced greater benefits from ICIs compared to those with other mutations. Steroid treatment for irAE does not interfere with the ORR or ICI response. Co-existing mutations of *STK11* with *KRAS* and *TP53* with *MET* mutations were negatively correlated with ICI response.

## Figures and Tables

**Figure 1 cancers-15-05450-f001:**
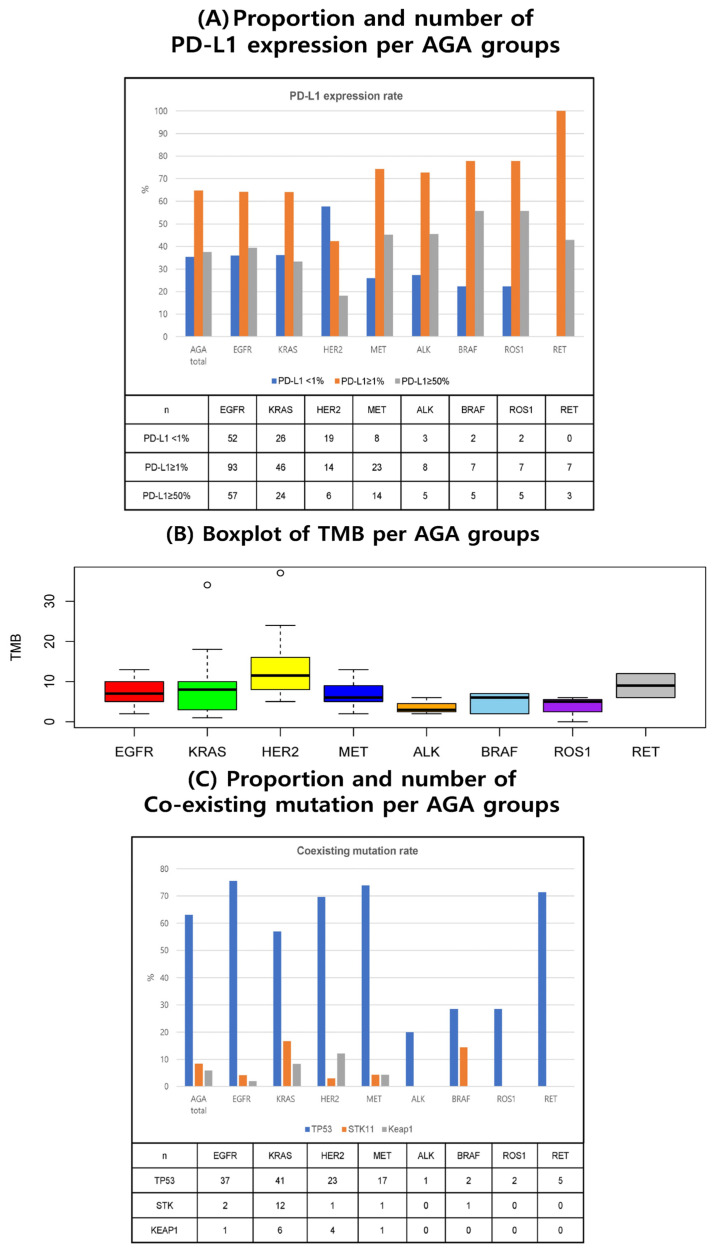
Molecular biomarker status according to AGA subgroups. AGA actionable genetic alteration; TMB tumor mutational burden.

**Figure 2 cancers-15-05450-f002:**
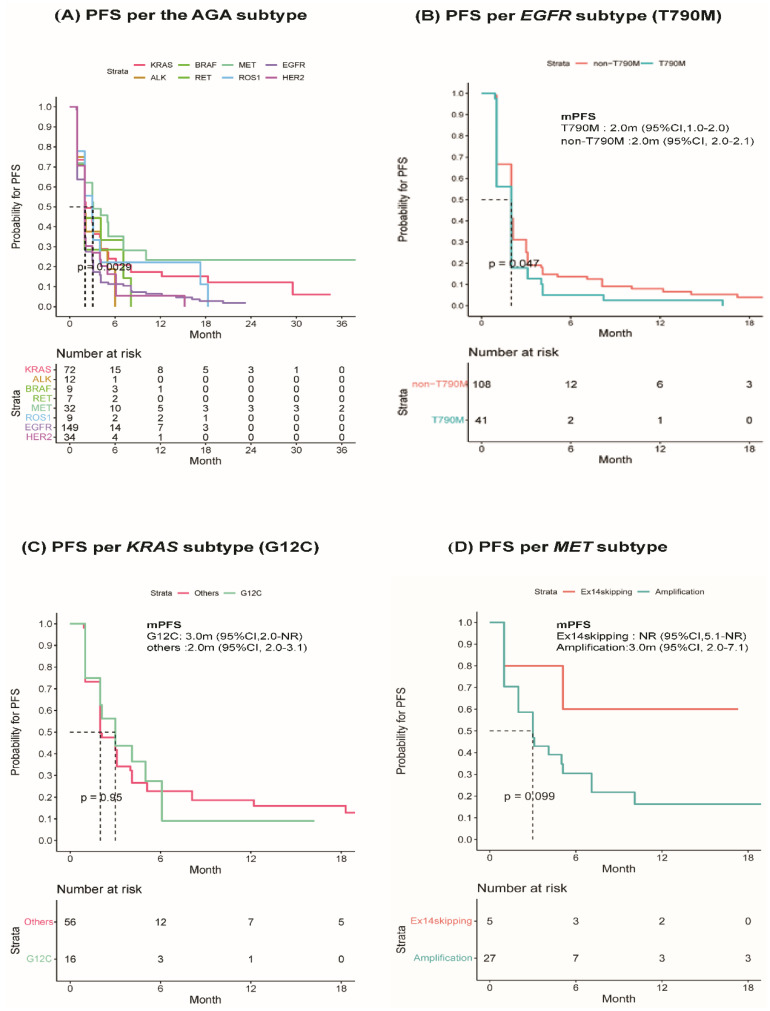
ICI response according to AGA subtypes.

**Figure 3 cancers-15-05450-f003:**
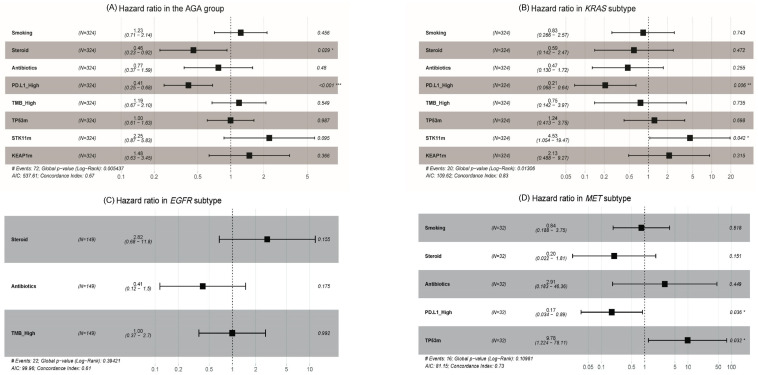
Hazard ratios of predictive biomarkers of ICI benefits per AGA subtype. AGA actionable genetic alteration; PD.L1_High, PD-L1 ≥ 1%; TMB_High, TMB ≥ 10 muts/mb; *TP53*m, *TP53*mutation positive; *STK11*m, *STK11* mutation positive; *KEAP1*m, *KEAP1* mutation positive. *: statistically significant, * *p* < 0.05; ** *p* < 0.01; *** *p* < 0.001.

**Figure 4 cancers-15-05450-f004:**
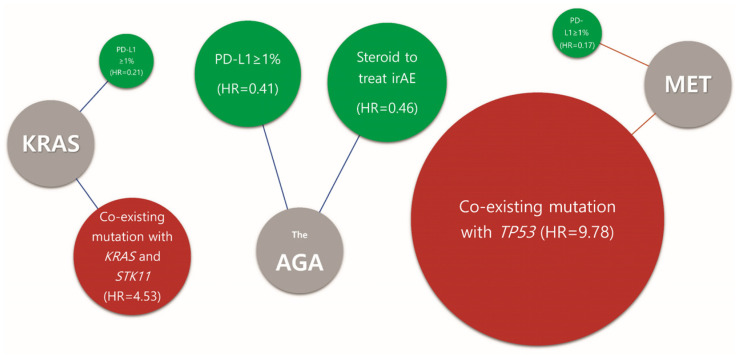
Algorithm for positive and negative predictive biomarkers associated with effective immunotherapy. Green: positive predictive marker. Red: negative predictive marker. Gray: Name of AGA sub group. The size of the circles represents the difference in HR.

**Table 1 cancers-15-05450-t001:** Clinical characteristics of patients with AGAs.

AGA Type n (%)		EGFR	KRAS	HER2	MET	ALK	BRAF	ROS1	RET	*p*
		n = 149 (46.0)	n = 72 (22.2)	n = 34 (10.5)	n = 32 (9.9)	n = 12 (3.7)	n = 9 (2.8)	n = 9 (2.8)	n = 7 (2.2)	
Age 1 (years)	Median age (95% range)	62.2 (39.1–81.7)	67.2 (44.4–81.7)	66.1 (47.3–76.5)	64.3 (44.1–78.4)	60.1 (53.8–82.0)	60.1 (54.0–73.60)	62.4 (44.4–80.9)	59.7 (30.8–65.9)	
	<65	90 (60.4)	30 (41.7)	17 (50.0)	18 (56.2)	9 (75.0)	7 (77.8)	7 (77.8)	5 (71.4)	0.06
	≥65	59 (39.6)	42 (58.3)	17 (50.0)	14 (43.8)	3 (25.0)	2 (22.2)	2 (22.2)	2 (28.6)	
Sex	Male	70 (47.0)	41 (56.9)	22 (64.7)	24 (75.0)	7 (58.3)	5 (55.6)	2 (22.2)	1 (14.3)	*0.01*
	Female	79 (53.0)	31 (43.1)	12 (35.3)	8 (25.0)	5 (41.7)	4 (44.4)	7 (77.8)	6 (85.7)	
Smoking	Never	90 (60.4)	34 (47.2)	13 (38.2)	11 (34.4)	6 (50.0)	5 (55.6)	8 (88.9)	7 (100)	*0.002*
	Ex or Current	59 (39.6)	38 (52.8)	21 (61.8)	21 (65.6)	6 (50.0)	4 (44.4)	1 (11.1)	0 (0.0)	
ECOG	0–1	147 (98.7)	67 (93.1)	33 (97.1)	31 (96.9)	12 (100)	9 (100)	9 (100)	7 (100)	0.45
	≥2	2 (1.3)	5 (6.9)	1 (2.9)	1 (3.1)	0 (0.0)	0 (0.0)	0 (0.0)	0 (0.0)	
Histology	Non-squamous	139 (93.3)	65 (90.3)	27 (79.4)	26 (81.2)	10 (83.3)	8 (88.9)	9 (100)	6 (85.7)	0.19
	Others	10 (6.7)	7 (9.7)	7 (20.6)	6 (18.8)	2 (16.7)	1 (11.1)	0 (0.0)	1 (14.3)	
Line of Therapy	2nd line	19 (12.8)	61 (84.7)	3 (25.0)	29 (90.6)	3 (25.0)	4 (44.4)	5 (55.6)	6 (85.7)	<0.001
	Later line	130 (87.2)	11 (15.3)	9 (75.0)	3 (9.4)	9 (75.0)	5 (55.6)	4 (44.4)	1 (14.3)	
Concomitant disease and therapy										
HTN	No	94 (63.1)	42 (58.3)	21 (61.8)	17 (53.1)	8 (66.7)	3 (33.3)	4 (44.4)	4 (57.1)	0.82
	Yes	54 (36.2)	30 (41.7)	13 (38.2)	14 (43.8)	4 (33.3)	6 (66.7)	5 (55.6)	3 (42.9)	
	NI	0 (0.0)	0 (0.0)	0 (0.0)	1 (3.1)	0 (0.0)	0 (0.0)	0 (0.0)	0 (0.0)	
DM	No	118 (79.2)	55 (76.4)	21 (61.8)	23 (71.9)	9 (75.0)	4 (44.4)	8 (88.9)	6 (85.7)	0.14
	Yes	31 (20.8)	17 (23.6)	13 (38.2)	8 (25.0)	3 (25.0)	5 (55.6)	1 (11.1)	1 (14.3)	
	NI	0 (0.0)	0 (0.0)	0 (0.0)	1 (3.1)	0 (0.0)	0 (0.0)	0 (0.0)	0 (0.0)	
RT *	No	130 (87.2)	64 (88.9)	25 (73.5)	29 (90.6)	10 (83.3)	9 (100)	7 (77.8)	6 (85.7)	0.35
	Yes	19 (12.8)	8 (11.1)	9 (26.5)	3 (9.4)	2 (16.7)	0 (0.0)	2 (22.2)	1 (14.3)	
Steroid ^†^	No	113 (75.8)	55 (76.4)	24 (70.6)	26 (81.2)	8 (66.7)	9 (100)	7 (77.8)	6 (85.7)	0.68
	Yes	36 (24.2)	17 (23.6)	10 (29.4)	6 (18.8)	4 (33.3)	0 (0.0)	2 (22.2)	1 (14.3)	
Antibiotics ^$^	No	126 (84.6)	56 (77.8)	31 (91.2)	29 (90.6)	7 (58.3)	9 (100)	9 (100)	6 (85.7)	0.05
	Yes	23 (15.4)	16 (22.2)	3 (8.8)	3 (9.4)	5 (41.7)	0 (0.0)	0 (0.0)	1 (14.3)	

Note: Values are shown as number (%) unless indicated otherwise. * including ccrt; ^†^ steroid use to treat irAE during ICI therapy; ^$^ antibiotic use. during ICI therapy; ICI, immune checkpoint inhibitor; AGA, actionable genetic alteration; ECOG, Eastern Cooperative Oncology Group performance status; HTN, hypertension; DM, diabetes mellitus; RT, radiotherapy.

**Table 2 cancers-15-05450-t002:** Molecular biomarker status.

Biomarkers N (%)		AGA n = 324	Wild Type n = 602	*p*	EGFR n = 149	KRAS n = 72	HER2 n = 34	MET n = 32	ALK n = 12	BRAF n = 9	ROS1 n = 9	RET n = 7	*p*
PD-L1 expression		317 (97.8)	596 (99.0)		145 (97.3)	72 (100)	33 (97.1)	31 (96.9)	11 (91.7)	9 (100)	9 (100)	7 (100)	
Median (95% CI)		10 (0–100)	9 (0–100)		10 (0–90)	7.5 (0–100)	0 (0–82)	10 (0–100)	25 (0–90)	50 (0–98)	50 (0–94)	10 (1.6–90)	
Cutoff	H (≥1%)	205 (64.7)	355 (59.6)	0.15	93 (64.1)	46 (63.9)	14 (42.4)	23 (74.2)	8 (72.7)	7 (77.8)	7 (77.8)	7 (100)	0.05
	L (<1%)	112 (35.3)	241 (40.4)		52 (35.9)	26 (36.1)	19 (57.6)	8 (25.8)	3 (27.3)	2 (22.2)	2 (22.2)	0 (0.0)	
TMB		92 (28.4)	92 (15.3)		22 (14.8)	29 (40.3)	16 (47.1)	12 (37.5)	3 (25.0)	5 (55.6)	3 (33.3)	2 (28.6)	
Median (95% CI)		7.0 (1.3–22.6)	10.5 (2.3–40.1)		7.0 (2.5–12.5)	8.0 (1.0–22.8)	11.5 (5.0–32.13)	6.0 (2.3–11.9)	3.0 (2.1–5.9)	6.0 (2.0–7.0)	5.0 (0.3–6.0)	9.0 (6.2–11.9)	
Cutoff	H (≥10)	26 (28.3)	49 (53.3)	0.001	6 (27.3)	7 (24.1)	11 (68.8)	1 (8.3)	0 (0.0)	0 (0.0)	0 (0.0)	1 (50.0)	0.01
Muts/Mb	L (<10)	66 (71.7)	43 (46.7)		16 (72.7)	22 (75.9)	5 (31.2)	11 (91.7)	3 (100)	5 (100)	3 (100)	1 (50.0)	
TP53		203 (62.7)	208 (34.6)		49 (32.9)	72 (100)	33 (97.1)	23 (71.9)	5 (41.7)	7 (77.8)	7 (77.8)	7 (100)	
	Mutation (+)	128 (63.1)	165 (79.3)	<0.001	37 (75.5)	41 (56.9)	23 (69.7)	17 (73.9)	1 (20.0)	2 (28.6)	2 (28.6)	5 (71.4)	0.01
	Mutation (−)	75 (36.9)	43 (20.7)		12 (24.5)	31 (43.1)	10 (30.3)	6 (26.1)	4 (80.0)	5 (71.4)	5 (71.4)	2 (28.6)	
STK11		202 (62.3)	208 (34.6)		49 (32.9)	72 (100)	33 (97.1)	23 (71.9)	4 (41.7)	7 (77.8)	7 (77.8)	7 (100)	
	Mutation (+)	17 (8.4)	25 (12.0)	0.30	2 (4.1)	12 (16.7)	1 (3.0)	1 (4.3)	0 (0.0)	1 (14.3)	0 (0.0)	0 (0.0)	0.13
	Mutation (−)	185 (91.6)	183 (88.0)		47 (95.9)	60 (83.3)	32 (97.0)	22 (95.7)	4 (100)	6 (85.7)	7 (100)	7 (100)	
KEAP 1		202 (62.3)	208 (34.6)		49 (32.9)	72 (100)	33 (97.1)	23 (71.9)	4 (41.7)	7 (77.8)	7 (77.8)	7 (100)	
	Mutation (+)	12 (5.9)	26 (12.5)	0.03	1 (2.0)	6 (8.3)	4 (12.1)	1 (4.3)	0 (0.0)	0 (0.0)	0 (0.0)	0 (0.0)	0.54
	Mutation (−)	190 (94.1)	182 (87.5)		48 (98.0)	66 (91.7)	29 (87.9)	22 (95.7)	4 (100)	7 (100)	7 (100)	7 (100)	

Note: Values are shown as number (%) unless indicated otherwise. Abbreviations: AGA, actionable genetic alterations; CI, confidence interval; TMB, tumor mutation burden. Wildtype used as the control.

**Table 3 cancers-15-05450-t003:** ICI response in receiving ICI monotherapy as second-line or later-line treatment.

		AGA	Wild Type		EGFR	KRAS	HER2	MET	ALK	BRAF	ROS1	RET
		n = 324	n = 602	*p*	n = 149	n = 72	n = 34	n = 32	n = 12	n = 9	n = 9	n = 7
Best response												
	CR	1 (0.3)	6 (1.0)		0 (0.0)	0 (0.0)	0 (0.0)	0 (0.0)	0 (0.0)	0 (0.0)	1 (11.1)	0 (0.0)
	PR	44 (13.6)	128 (21.3)		14 (9.4)	16 (22.2)	1 (2.9)	8 (25.0)	1 (8.3)	1 (11.1)	1 (11.1)	2 (28.6)
	SD	75 (23.1)	147 (24.4)		23 (15.4)	23 (31.9)	10 (29.4)	7 (21.9)	4 (33.3)	3 (33.3)	2 (22.2)	3 (42.9)
	PD	192 (59.3)	303 (50.3)		105 (70.5)	32 (44.4)	21 (61.8)	16 (50.0)	6 (50.0)	5 (55.6)	5 (55.6)	2 (28.6)
	NE	12 (3.7)	18 (3.0)		7 (4.7)	1 (1.4)	2 (5.9)	1 (3.1)	1 (8.3)	0 (0.0)	0 (0.0)	0 (0.0)
ORR	CR + PR	45 (13.9)	134 (22.3)	0.82	14 (9.4)	16 (22.2)	1 (2.9)	8 (25.0)	1 (8.3)	1 (11.1)	2 (22.2)	2 (28.6)
PFS	Median (95% CI) (months)	2.0 (2.0–2.0)	2.1 (2.0–3.0)	<0.001	2.0 (2.00–2.03)	2.1 (2.0–3.1)	2.0 (2.0–3.0)	3.1 (2.0–10.1)	2.0 (2.0-NR)	2.0 (2.0-NR)	3.0 (2.0-NR)	2.0 (1.0-NR)
OS	Median (95% CI) (months)	12.2 (10.1–15.3)	10.1 (8.2–11.2)	0.06	9.2 (7.1–13.2)	12.2 (9.1–22.3)	10.1 (8.1-NR)	22.3 (11.2-NR)	5.0 (3.0-NR)	11.1 (6.0-NR)	NR (34.6-NR)	27.3 (18.3-NR)
The 12-month PFS rate		11.3% (8.1–15.7)	18.2% (15.2–21.8)		6.4% (3.35–12.38)	17.4% (10.19–29.57)	5.4% (0.89–32.66)	23.5% (11.91–46.20)	0% NA	22.2% (6.55–75.44)	22.2% (6.55–75.44)	0% NA

Note: Values are shown as number (%) unless indicated otherwise, immune checkpoint inhibitor; AGA, actionable genetic alterations; CR, complete response; PR, partial response; SD, stable disease; PD, progressive disease; NE, not evaluable; ORR, overall response rate; PFS, progression-free survival; OS, overall survival; CI: confidence interval; NR: not reached; NA: not applicable. Wildtype as the control.

## Data Availability

The data presented in this study are available upon approval from the Data Review Board of Samsung Medical Center.

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
