# Peer review of "Real-World Outcomes of Immunotherapy in Second- or Later-Line Non-Small Cell Lung Cancer with Actionable Genetic Alterations"

_cancers, 2023, doi:10.3390/cancers15225450_

Round 1

Reviewer 1 Report

Comments and Suggestions for Authors

This study is interesting with clinical significance. Immunotherapy has revolutionized the tumor therapy, especially in Non-Small Cell Lung Cancer. The authors put forward a new and comprehensive point of view on of immune checkpoint inhibitors treatment across AGA subtypes. The followings are some comments to the authors.

Comments:

1. PD-L1 expression is an important factor affecting immunotherapy, I suggest that the PFS should be analyzed according PD-L1 1% and <1%.

2. There was a large difference in sample size between the subgroups in Figure 2(A)-(D), how to explain the effect of the unbalanced sample size on the results? Please state that in “3.3. Treatment Outcomes of Immune Checkpoint Inhibitors”.

3. How many patients were previously treated with ICI? What was the effect of previous ICI therapy? For example, the proportion of patients with time from the last dose to disease progression 6 months. Because previous treatment may have an impact on later line therapy.

4. What is the proportion of patients with 2 line treatment? Please state that in Table 1.

Author Response

We would like to express our deep appreciation for your time to review this manuscript. Please find the detailed responsed below and the corresponding track changes in the re-submitted files.  

1. PD-L1 expression is an important factor affecting immunotherapy, I suggest that the PFS should be analyzed according PD-L1≥1% and <1%

REPLY: Thank you for the constructive feedback on our manuscript. We hope this revision has appropriately addressed the comments. In the result section, we described the PFS of the AGA group, KRAS, EGFR, HER2, and MET according to PD-L1 High (≥1%) and Low (<1%) group with corresponding Table S4 in the supplementary.

Changes in the manuscript (line of tracked version): Manuscript Page 8, lines 218-221, Supplementary document Page 5, lines 25-27 for Table S4. We added Among the AGA subgroups, the analysis of the impact of PD-L1 level on immunotherapy identified a significant difference in mPFS between High (≥1%) and Low (<1%) (4.0 months; 95%CI 2.1-6.1 vs 2.0 month; 95%CI 1.0-3.0, p<0.001) in the KRAS subgroup (Supplementary Table S4).

 Table S4. PFS analysis of PD-L1 expression in patients with AGA mutations (EGFR, KRAS, HER2, and MET subgroups).

Survival Analysis

AGA
n=324

KRAS
n=72

EGFR
n=149

HER2
n=34

MET 
n=32

PFS

PFS

PFS

PFS

PFS

n (%)

median (95%CI)

p

n (%)

median (95%CI)

p

n (%)

median (95%CI)

p

n (%)

median (95%CI)

p

n (%)

median (95%CI)

p

Clinical factor

324

72

149

34

32

PD-L1 expression

ï¼´

72

145

33

31

cutoff 1%

H (≥1%)

   205 (64.7)

2.0 (2.0-3.0)

<0.001

46 (63.9)

4.0 (2.1-6.1)

<0.001

93 (64.1)

2.0 (2.0-2.0)

0.2

14 (42.4)

2.0 (2.0-NR)

0.1

23 (74.2)

5.1 (2.0-NR)

0.2

L (<1%)

112 (35.3)

2.0 (2.0-2.0)

26 (36.1)

2.0 (1.0-3.0)

52 (35.9)

2.0 (2.0-2.0)

19 (57.6)

2.0 (1.0-3.0)

8 (25.8)

2.5 (1.0-NR)

2. There was a large difference in sample size between the subgroups in Figure 2(A)-(D), how to explain the effect of the unbalanced sample size on the results? Please state that in “3.3. Treatment Outcomes of Immune Checkpoint Inhibitors”.

REPLY: Thank you for your comment. We thought the unbalanced sample sizes in Figure 2(A)-(D) did not impact the results for the following reasons.

Firstly, Figure 2 was not designed for direct comparison. Figure 2A presents an integrated analysis of all eligible patients across AGA subtypes, comparing survival rates between groups. Subsequently, Figures 2(B), 2(C), and 2(D) concentrate on three mutation groups (EGFR, KRAS, and MET) characterized by high expression rates to enhance a better understanding of AGA subtypes. In particular, further scrutiny was applied to analyze survival rates within each group, with a focus on T790M of EGFR (Fig 2B), G12C of KRAS (Fig 2C), and exon 14 skipping MET (Fig 2D).

Changes in the manuscript (line of tracked version): not applicable.

3. How many patients were previously treated with ICI? What was the effect of previous ICI therapy? For example, the proportion of patients with time from the last dose to disease progression ≥6 months. Because previous treatment may have an impact on later-line therapy.

REPLY: Thank you for your comment. This study exclusively targeted patients treated in 2nd and later-line, as the impact of previous ICI therapy is notably restricted in Korea. This limitation arises from the fact that Tyrosine Kinase Inhibitors (TKIs) are the standard of care (SOC) in the first line for patients with EGFR and ALK mutations, as per the NCCN and the Korean National Health Insurance guideline. Additionally, other mutations have a limited patient number within the AGA cohort because they have been covered by the Korean National Health Insurance since March 2022.

Changes in the manuscript (line of tracked version): not applicable.

4. What is the proportion of patients with 2nd line treatment? Please state that in Table 1.

REPLY: Thank you for pointing this out. In this study, out of the 324 patients who treated 2nd and later-line treatments, 47.2% (153 patients) were identified as patients who received ICI in the 2nd line therapy. We described the result of the proportion of patients with 2nd line treatment with corresponding Table 1 and Table S1.

Changes in the manuscript (line of tracked version): Insert sentence in the result section of page 4 Lines 147, 148-149, and Table 1, line 154 in the Manuscript, and page 2, Table S1, Line 13 in a supplementary document.

Regarding the clinical characteristics, there were no significant differences except in sex (p = 0.01) smoking status (p = 0.002) and line of therapy (p<0.001) across the AGA subtypes. In total, 46.9% (n = 152) of the patients were female and 46.3% (n = 150) were ex- or current smokers, 47.2% (n=153) were identified as patients received ICI in the 2nd line therapy in the AGA group.

Table 1. Clinical Characteristics of Patients WITH AGAs.

AGA type n (%)

EGFR

KRAS

HER2

MET

ALK

BRAF

ROS1

RET

p

n = 149 (46.0)

n = 72 (22.2)

n = 34 (10.5)

n = 32(9.9)

n = 12 (3.7)

n = 9 (2.8)

n = 9 (2.8)

n = 7 (2.2)

Demographic data

Line of Therapy

2nd line

19 (12.8)

61 (84.7)

3 (25.0)

29 (90.6)

3 (25.0)

4 (44.4)

5 (55.6)

6 (85.7)

<0.001

Later line

130 (87.2)

11 (15.3)

9 (75.0)

3 (9.4)

9 (75.0)

5 (55.6)

4 (44.4)

1 (14.3)

Table S1. Clinical characteristics of patients receiving ICI monotherapy as second-line or later-line treatment.

AGA type n (%)

AGA

Wild Type

p

n = 324

n = 602

Demographic data

Line of Therapy

2nd line

153 (47.2)

524 (87.0)

<0.001

Later line

171 (52.8)

78 (13.0)

Reviewer 2 Report

Comments and Suggestions for Authors

In this study, the authors reviewed their experience with the efficacy of immune checkpoint inhibitors (ICIs) in treating non-small cell lung cancer (NSCLC) patients with various actionable genetic alterations (AGAs). As expected, the overall result is modest, while some patients with BRAF mutations demonstrate improved survival. as known, patients with nsclc harboring co-existing mutation of STK11 with KRAS mutation (HR=4.53) and TP53 with MET mutation (HR=9.78) was negatively associated with survival.

T the end, the efficacy of ICI treatment varied across AGA subtypes, but patients with KRAS, MET, and BRAF mutations demonstrated relatively long-duration benefits of ICI therapy. PD-L1 was a significant positive predictive biomarker in all AGA groups.

I would simply suggest the authors to consider to elaborate and insert a sort of concise algorithm including favorable predictive molecular fingings associated with effective immunotherapy and the opposite, that should be greatly appreciated by readers.

Comments on the Quality of English Language

None

Author Response

In this study, the authors reviewed their experience with the efficacy of immune checkpoint inhibitors (ICIs) in treating non-small cell lung cancer (NSCLC) patients with various actionable genetic alterations (AGAs). As expected, the overall result is modest, while some patients with BRAF mutations demonstrate improved survival. as known, patients with nsclc harboring co-existing mutation of STK11 with KRAS mutation (HR=4.53) and TP53 with MET mutation (HR=9.78) was negatively associated with survival.

At the end, the efficacy of ICI treatment varied across AGA subtypes, but patients with KRAS, MET, and BRAF mutations demonstrated relatively long-duration benefits of ICI therapy. PD-L1 was a significant positive predictive biomarker in all AGA groups.

I would simply suggest the authors to consider to elaborate and insert a sort of concise algorithm including favorable predictive molecular findings associated with effective immunotherapy and the opposite, that should be greatly appreciated by readers.

REPLY: Thank you for your insightful suggestion. As reviewer’s suggestion, we developed a graphical algorithm that incorporates favorable and unfavorable predictive molecular findings, distinguished by different colors and sizes in the result section.

Changes in the manuscript (line of tracked version): Manuscript page 9, lines 245-248
